# International and regional spread of carbapenem-resistant *Klebsiella pneumoniae* in Europe

Mabel Budia-Silva[1], Tomislav Kostyanev[2,3], Stefany Ayala-Montaño[1], Jose Bravo-Ferrer Acosta [4], Maria Garcia-Castillo[5,6], Rafael Cantón [5,6], Herman Goossens[2], Jesus Rodriguez-Baño [4,6], Hajo Grundmann [1] & Sandra Reuter [1] ✉

Carbapenem-resistant *Klebsiella pneumoniae* (CRKP) are of particular concern due to the spread of antibiotic resistance genes associated with mobile genetic elements. In this study, we collected 687 carbapenem-resistant strains recovered among clinical samples from 41 hospitals in nine Southern European countries (2016-2018). We identified 11 major clonal lineages, with most isolates belonging to the high-risk clones ST258/512, ST101, ST11, and ST307. *bla*$_{KPC-like}$ was the most prevalent carbapenemase-encoding gene (46%), with *bla*$_{OXA-48}$ present in 39% of isolates. Through the combination and comparison of this EURECA collection with the previous EuSCAPE collection (2013-2014), we investigated the spread of high-risk clones circulating in Europe exhibiting regional differences. We particularly found *bla*$_{KPC-like}$ ST258/512 in Greece, Italy, and Spain, *bla*$_{OXA-48}$ ST101 in Serbia and Romania, *bla*$_{NDM}$ ST11 in Greece, and *bla*$_{OXA-48-like}$ ST14 in Türkiye. Genomic surveillance across Europe thus provides crucial insights for local risk mapping and informs necessary adaptions for implementation of control strategies.

Gram-negative *Klebsiella pneumoniae* (*Kp*) are part of the microbiota in humans and are considered opportunistic pathogens able to cause hospital- and community-acquired infections such as pneumonia, bloodstream and urinary tract infections[1]. Clinically, these infections are treated with fluoroquinolones, aminoglycosides, cephalosporins, and, as a last resort, carbapenem antibiotics. However, *Kp* may become resistant to carbapenems mainly through the acquisition of resistance genes encoding carbapenemases, or the production of extended-spectrum beta-lactamases or cephalosporinases combined with porin alterations[2]. Carbapenemase genes are of particular concern as they can spread in association with mobile genetic elements (MGE) that are

part of plasmids and transposons. These carbapenemase genes, mainly *bla*$_{KPC}$, *bla*$_{NDM}$, *bla*$_{VIM}$, and *bla*$_{OXA-48}$, are often associated with particular and successful nosocomial clones, sometimes with the close relationship between a lineage and the antibiotic resistance determinants[3].

The increase of carbapenem-resistant *Kp* (CRKP) represents an unquestionable threat to the health of hospitalized patients globally, with a high mortality rate compared to patients infected with carbapenem-susceptible *Kp*[4,5]. Indeed, CRKP is a relevant public health problem with economic effects[6], and it was recognized as a critical pathogen by the World Health Organization (WHO) for the need of

[1]Institute for Infection Prevention and Control, University of Freiburg – Medical Center, Freiburg, Germany. [2]Laboratory of Medical Microbiology, University of Antwerp, Antwerp, Belgium. [3]Research Group for Global Capacity Building, National Food Institute, Technical University of Denmark, Kgs. Lyngby, Denmark. [4]Unidad Clínica de Enfermedades Infecciosas y Microbiología, Instituto de Biomedicina de Sevilla (IBiS)/CSIC, Hospital Universitario Virgen Macarena; and Departamento de Medicina, Universidad de Sevilla, Seville, Spain. [5]Servicio de Microbiología, Hospital Universitario Ramón y Cajal and Instituto Ramón y Cajal de Investigación Sanitaria (IRYCIS), Madrid, Spain. [6]CIBER de Enfermedades Infecciosas (CIBERINFEC), Institute de Salud Carlos III, Madrid, Spain. ✉e-mail: sandra.reuter@uniklinik-freiburg.de

new antimicrobials[7]. Different surveillance programs are collaboratively running at several hospitals in many countries. In Europe, national outbreaks of CRKP have been reported mainly in Southern European countries such as Greece, Spain, and Italy where also the highest prevalence is observed[8,9]. To limit the high incidence, increasing spread, and unraveling transmission pathways of CRKP, it is therefore relevant to study their characteristics, including distribution of lineages and resistance determinants at regional and local levels as well as internationally.

The COMBACTE consortium pursues the prevention and treatment of antibiotic-resistant-associated infections through four main projects. Among them, COMBACTE-CARE, seeks to support the development of new treatment options, together with the analysis of clinical and epidemiological datasets in all European member states and affiliated countries[10,11]. As part of this, the European prospective cohort study on *Enterobacterales* showing resistance to carbapenems (EURECA) aimed to understand how the patients across Europe are infected and currently treated for *Enterobacterales*-associated infections, but also which subgroups of patients responded well to different antibiotic treatments[10]. Local laboratories submitted carbapenem non-susceptible isolates to EURECA from May 2016 to November 2018 from cohorts of patients with bacterial infections in Southern European countries. Our task was to characterize and identify circulating clones of CRKP by analyzing 687 genomes. In addition, we contextualized the spread of CRKP for a broader population view by comparing our data to the previous EuSCAPE study[8], which included the carbapenem-susceptible and non-susceptible isolates during 2013–2014 sample from a wider range of European countries.

## Results

### Presence of carbapenemase genes across different ST

In the EURECA collection, 683 isolates were classified by the phylogenetic analysis into *Klebsiella pneumoniae* sensu stricto, with two isolates each in *Klebsiella variicola* and *Klebsiella quasipneumoniae*. Of all the *Kp* sensu stricto, we identified 50 different sequence types (STs) and nine novel single locus variants (SLV) (Supplementary Data 1). Most isolates ($n = 599$, 87%) were grouped in one of 11 clonal lineages and their single-locus variants (Table 1), with ST258/512 being the dominant lineage ($n$=204; 30%), and ST512 the main single ST ($n = 116$, 17%) (Fig. 1). Other clonal lineages in order of prevalence were ST11 ($n = 116$, 17%), ST101 ($n = 87$, 15%), ST307 ($n = 71$, 10%), ST15 ($n = 42$, 6%), and ST147 ($n = 40$, 6%). Of the carbapenemase genes, $bla_{KPC-like}$ ($n = 314$, 46%) were the most abundant, mainly linked to ST258/512 (Fig. 1). $bla_{OXA-48-like}$ ($n = 266$, 39%) was the second most widespread gene found in several STs, for instance, ST15, ST101, ST11. $bla_{NDM-1}$ was found in 96 (14%) isolates either on its own or in combination with $bla_{OXA-48-like}$ genes, and interestingly, it was present in half of the ST11 isolates. A combination of two carbapenemase genes was found in 38 isolates (5.5%), with the majority of isolates having a combination of $bla_{OXA-48} + bla_{NDM-1}$ ($n = 25$, 66%) (Fig. 1, Table 1). Despite re-culturing under antibiotic selection, DNA extraction, and re-sequencing, we could not find a carbapenemase in 22 isolates. All of these were classed phenotypically as "non-KPC, non-metallo beta-lactamase". As we were unable to recover a carbapenemase, we assume that the resistance mechanism is something other than an acquired carbapenemase or that any plasmid conferring resistance was lost before sending the strain for sequencing.

### Geographic distribution of major clonal lineages and carbapenemase genes

We compared EURECA isolates to the EuSCAPE collection ($n = 1717$) recovered during 2013–2014 with a similar sampling framework (Fig. 2d)[8]. Carbapenem-susceptible isolates were only collected as part of the EuSCAPE collection, grouping into a variety of clonal lineages, with 27.2% ($n = 467$) not belonging to any of the major clonal lineages,

apart from Serbia where half of the isolates were ST101 (Fig. 2a). In contrast, carbapenem resistant isolates were grouped in eleven common clonal lineages in both collections with minor changes in the order of prevalence (87% EuSCAPE, 88% EURECA of each collection; Fig. 2b and c), except for the ST35 present only in EuSCAPE. The most prominent clonal lineage was ST258/512, then ST11 and ST101, with noticeable regional differences in abundance (Table 1, Supplementary Table 1). Italy was dominated by ST258/512 and associated with $bla_{KPC-like}$ genes (Fig. 2b and e). Whilst ST258/512 was present in Greece as well, the country had a higher diversity of clones (17 different STs, vs. 13 STs in Italy in EURECA) with ST11-$bla_{NDM}$ as prominent as ST258/512 (41 and 58 isolates, respectively; Fig. 2c, Table 1). In Eastern Europe, in Serbia and Romania, the abundant clone was ST101, although Serbia and Romania had a second dominant clone (ST11, and ST15 in EURECA, respectively; Fig. 2b and c). In the carbapenemases present, Serbia had a higher percentage of isolates harboring $bla_{NDM}$ gene in the EuSCAPE collection (44%, $n = 24$), whereas the later EURECA collection showed a dominance of $bla_{OXA-48}$ (75%, $n = 82$). In contrast, although numbers are small in the EURECA collection, Romania had a higher percentage of $bla_{NDM}$ isolates (35%, $n = 11$) compared to EuSCAPE (6%, $n = 4$) (Fig. 2e and f).

Other countries, such as Spain and Türkiye displayed a more mixed picture of different clonal lineages with several clones at near-equal proportions, yet changed over time. Though ST11, ST15, and ST147 still feature prominently in Spain (18%, 17%, 10%, respectively), ST258 is now dominant (28%), and 16% are ST307 (Fig. 2c). Türkiye showed a number of different lineages in the EuSCAPE collection, however in EURECA, ST14 is dominant (49%, $n = 24$; Fig. 2c). Common to Serbia, Spain and Türkiye is the presence of the $bla_{OXA-48}$ gene (Fig. 2e and f). As this gene is often carried on a promiscuous plasmid[12], this might explain the wide dispersal in various lineages.

### ST258/512 – the worldwide spread of a successful lineage

The dissemination of the ST258/512 has been characterized largely in previous studies. We, therefore, analyzed the EURECA collection (Fig. 3, highlighted in the first column in light blue; https://microreact.org/project/gUrL4jULY1JEPNExrFK4zH-globalcollectionclonallineage258512) together with 652 publicly available genomes around the world[13–16], including the EuSCAPE collection[8].

KPC-producing *K. pneumoniae* were first reported as outbreaks in the USA, initially focussed on the East Coast before spreading more widely[14,15,17]. From the oldest genomes reported of ST258, we can see that these originate from the USA (light green, Fig. 3), and that they also show considerable diversity, which has been previously described as "paraphyletic"[14]. Emergence of ST258 has been dated to the mid-1990s, before showing in clinical presentations and outbreaks at the beginning of the 2000s[14]. These ST258 isolates are mainly capsule-type KL107, carry the $bla_{KPC-3}$ variant, and have been predominantly isolated from the USA (Fig. 3)[14,15].

Clade 1 is the first major export of ST258 outside of the USA. This clade simultaneously changed both the capsule as well as the KPC variant, now KL106 and $bla_{KPC-2}$ (Supplementary Data 2)[14,16]. Previous reports[8,14] noted that isolates in this clade have a close association with Greece, with isolates either being isolated there or being associated with recent travel or healthcare exposure in Greece. This clone was first reported sporadically in Greece between 2006-2008[18]. This introduction and expansion in Greece is detected in the EURECA collection, with isolates from Serbia and Montenegro nested within the diversity of the Greek isolates.

Before the second major clade, previously labeled as clade 2, there are a handful of US isolates which are both KL107 and $bla_{KPC-2}$. Given the characteristics of the paraphyletic isolates, it would appear that clade 2 emerged directly out of this background population. Clade 2 of ST258 was introduced into Israel, with early reports from 2005, where also the emergence of the single locus variant ST512 was

**Table 1 | Characteristics of all submitted CRKP with different carbapenemase genes. Full data available in Supplementary Data 1**

| Clonal lineages | Total number of isolates (% of total) | Countries with more than 10 isolates | Number of different countries | STs with more than 10 isolates | Carbapenemase gene variants and number of isolates | | | Two carbapenemases | No carbapenemase found |
|---|---|---|---|---|---|---|---|---|---|
| | | | | | Class A | Class B | Class D | | |
| ST258/512 | 204 (30%) | Italy (n = 74) Greece (n=58) Spain (n=58) | 6 | ST512 (n = 116) ST258 (n = 73) ST1519 (n = 12) | $bla_{KPC-2}$ (n = 69) $bla_{KPC-3}$ (n = 134) $bla_{KPC-36}$ (n = 1) | $bla_{NDM-1}$ (n = 1) | | 1 | |
| ST11/437/340 | 116 (17%) | Greece (n = 41) Serbia (n = 37) Spain (n = 36) | 5 | ST11 (n = 78) ST437(n = 33) | $bla_{KPC-2}$ (n = 4) | $bla_{NDM-1}$ (n = 47) $bla_{VIM-1}$ (n = 1) | $bla_{OXA-48-like}$ (n = 60) | 3 | 7 |
| ST101 | 87 (15%) | Serbia (n = 56) Romania (n = 17) | 5 | ST101 (n = 83) | $bla_{KPC-2}$ (n = 2) $bla_{KPC-3}$ (n = 5) | $bla_{NDM-1}$ (n = 4) | $bla_{OXA-48}$ (n = 66) | 2 | 12 |
| ST307 | 71(10%) | Spain (n = 33) Italy (n = 24) | 5 | ST307 (n = 68) | $bla_{KPC-2}$ (n = 16) $bla_{KPC-3}$ (n = 27) | $bla_{NDM-1}$ (n = 4) $bla_{VIM-1}$ (n = 1) | $bla_{OXA-48}$ (n = 22) | 1 | 1 |
| ST15 | 42 (6%) | Spain (n = 34) | 5 | ST15 (n = 38) | $bla_{KPC-2}$ (n = 1) $bla_{KPC-3}$ (n = 1) | $bla_{NDM-1}$ (n = 2) $bla_{VIM-1}$ (n = 1) | $bla_{OXA-48-like}$ (n = 37) | 1 | 1 |
| ST147 | 40 (6%) | Spain (n = 20) Greece (n = 15) | 6 | ST147 (n = 35) | $bla_{KPC-2}$ (n = 12) $bla_{KPC-3}$ (n = 1) | $bla_{NDM-1}$ (n = 13) $bla_{VIM-1}$ (n = 6) | $bla_{OXA-48-like}$ (n = 21) | 14 | 1 |
| ST14 | 24 (4%) | Türkiye (n = 24) | 1 | ST14 (n = 20) | | $bla_{NDM-1}$ (n = 12) | $bla_{OXA-48-like}$ (n = 24) | 12 | |
| ST405 | 5 | | 2 | | $bla_{KPC-2}$ (n = 2) | | $bla_{OXA-48}$ (n = 3) | | |
| ST37 | 4 | | 3 | | | | $bla_{OXA-48}$ (n = 4) | | |
| ST45 | 4 | | 2 | | $bla_{KPC-3}$ (n = 2) | | $bla_{OXA-48}$ (n = 2) | | |
| ST17 | 2 | | 2 | | | | $bla_{OXA-48}$ (n = 2) | | |
| Other | 84 (12%) | Greece (n = 47) Serbia (n = 10) Türkiye (n = 10) | 7 | ST39 (n = 30) ST395(n = 14) | $bla_{KPC-2}$ (n = 33) $bla_{KPC-3}$ (n = 4) | $bla_{NDM-1}$ (n = 13) $bla_{VIM-1}$ (n = 14) | $bla_{OXA-48-like}$ (n = 25) | 5 | |

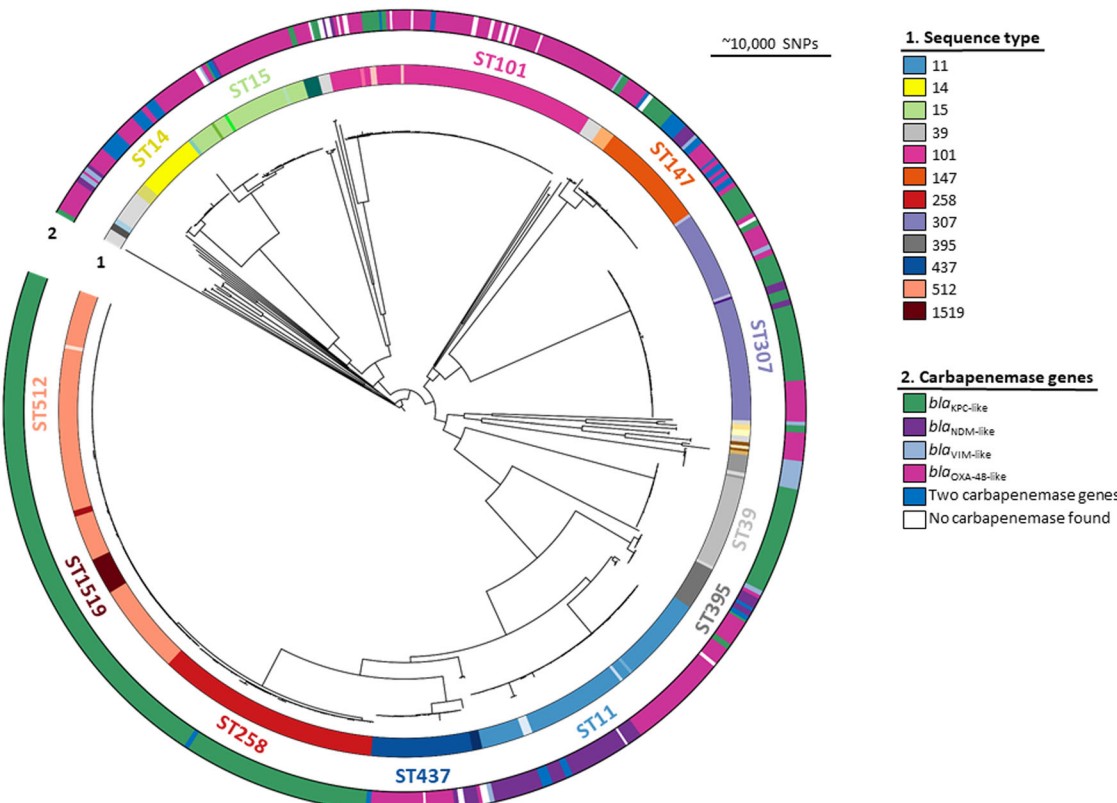

**Fig. 1 | Phylogenetic overview tree of the complete CRKP EURECA collection.** This phylogenetic tree contains 683 *Klebsiella pneumoniae* sensu stricto carbapenem resistant isolates collected as part of the EURECA study. The inner ring shows sequence type, and the different colors of the outer ring depict the carbapenemase genes. Main Sequence types with more than five isolates have been labeled.

documented[8,13,19]. Following this evolution and widespread dissemination within Israel, the clone was then introduced successfully into Italy (Fig. 3), likely in a single event[8]. Further spread of ST512 then occurred to Spain, which was not detected within the EUSCAPE dataset; this may be due to sampling in a different city than before (Fig. 3).

Microevolution of other single locus variants through mutations in *rpoB* is visible within the ST512 clade: ST868 (Fig. 3, red), has arisen on two independent occasions due to the same mutation in *rpoB*, making this ST not monophyletic. A new monophyletic ST1519 (orange) appears to be particularly prevalent in Bologna, as most isolates originate from that city.

Allelic variant mutations of $bla_{KPC}$ gene have been described that confer resistance to ceftazidime/avibactam. In this collection, however, we found only one ST1519 isolate carrying $bla_{KPC-36}$ (Fig. 3).

### ST307—the emergence of a new threat

Similarly to ST258/512, ST307 ($n = 68,10\%$) is an important globally distributed clone[20,21], therefore we analyzed the phylogenetic relationship of 113 ST307 isolates and three single locus variants (SLV) from the EuSCAPE and EURECA collections. This clone was frequently found in Greece, Italy, and Spain ($n = 81$, 70%; Table 1, Fig. 4). We investigated the previously proposed cutoff of 21 SNPs[8] across the different major clonal lineages, and found it to be comparable across all (Supplementary Fig. 1). We thus use this as a guiding value to identify putative within- or between-hospital clusters. In ST307, we found considerable diversity within this clone (Fig. 4), but found that isolates with less than 21 SNPs were indeed isolated from the same hospitals. This then proposed two small outbreaks in Spain within single hospitals (Fig. 4), one in which most isolates carried the $bla_{OXA-48}$ gene ($n = 10$), and a second in which $bla_{KPC-3}$ ($n = 7$; $n = 1$ combination with $bla_{OXA-162}$) was found (Fig. 4 purple boxes). Further putative

transmission events using this cut-off were detected in Greece and Italy (orange branches). In Italy, we also found a closely related clade (pink box) composed of both collections as well as different hospitals, indicating that this had been circulating for several years and is more widely disseminated. Isolates carried either $bla_{KPC-2}$, $bla_{KPC-3}$, or no carbapenemase, indicating fluidity with respect to high-level resistance. Overall, in this European collection, the most prevalent carbapenemase genes were $bla_{KPC-like}$, particularly in Italy and Greece, which also corresponded to the most prevalent carbapenemase type in these countries (Figs. 2b and 4).

In a global context, including EuSCAPE and EURECA collections as well as globally available isolates, the phylogenetic analysis showed a high diversity, with multiple small clades (Supplementary Fig. 2, https://microreact.org/project/rFyHR2a3iN1FDh7kJKj8pf-clonallineagest307globalcollectionproject). Two larger clades from the USA and Zambia stand out, particularly as they do not carry carbapenemases. Apart from these, the smaller clades are often country-specific, for example for Italy, Spain, France, and Germany. In ST307, various carbapenemases were found, alone and in combinations. ST258/512 shows a similar global diversity in average SNP distances between isolates as ST307 (116 SNPs and 110 SNPs, respectively). In contrast to ST258/512, ST307 shows no single carbapenemase or sublineage that is predominantly globally distributed.

### ST11 and ST101—the importance of knowing local conditions

A total of 330 isolates of ST11 and the single locus variants ST340 and ST437 from the EuSCAPE and EURECA collections were analyzed. The phylogenetic tree formed three clades (Fig. 5a): clade 1 containing the isolates belonging exclusively to ST437, clade 2 with ST11 isolates plus a nested ST340 clade, and clade 3 with ST11 isolates only. The

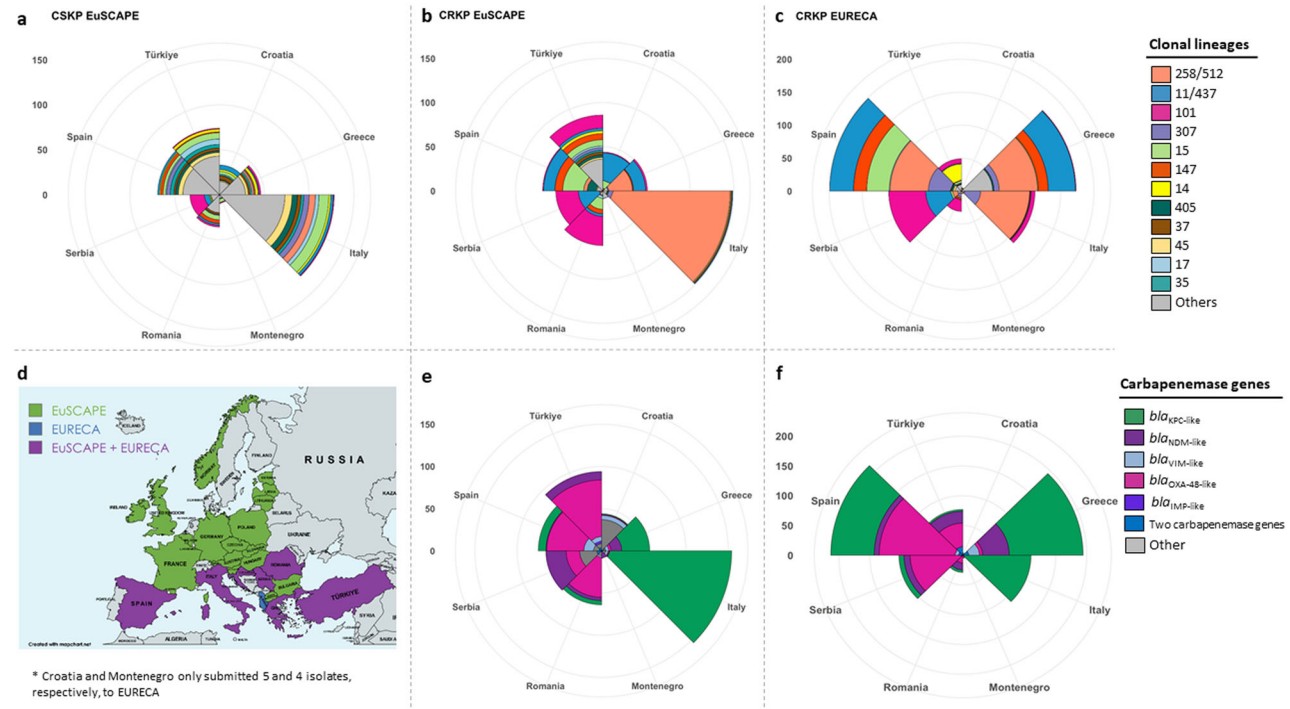

**Fig. 2 | Comparison of EuSCAPE and EURECA surveys of carbapenem resistant _K.pneumoniae._ a** Proportion of the EuSCAPE carbapenem susceptible (CSKP) clonal lineages. **b** and **c** Clonal lineages of CRKP isolates of EuSCAPE and EURECA, respectively. **d** European countries, which participated in EuSCAPE (green), EURECA (blue), and the overlap of both surveys (purple). Map created with map-chart.net and included under CC BY-SA 4.0 licence. **e** and **f** Distribution of carbapenem resistance genes in EuSCAPE and EURECA, respectively.

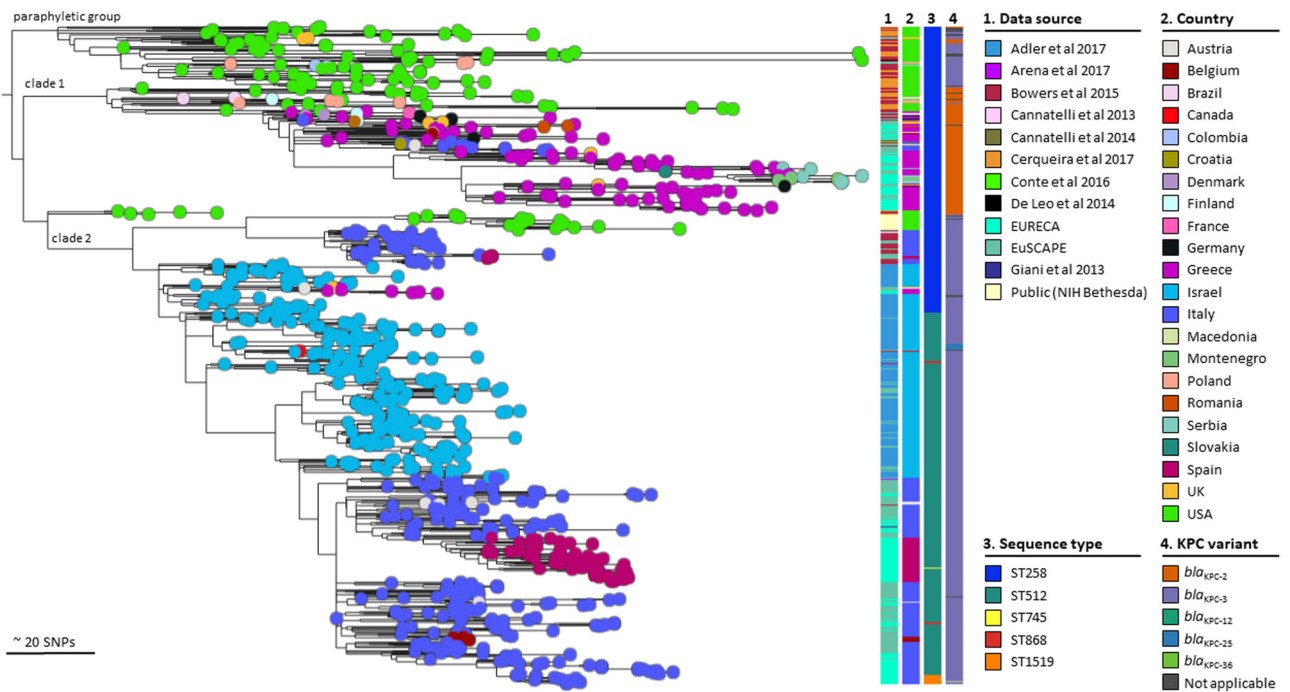

**Fig. 3 | Global spread of the epidemic ST258/512.** A phylogenetic tree of 856 isolates of ST258, ST512 and single locus variants with 204 isolates from EURECA, 236 isolates submitted to EuSCAPE, and 415 isolates with publicly available sequence data. Reference genome: 30660/NJST258 (CP006923). Tree tip colors correspond to country. Columns indicate: Project, (**1**) country, (**2**) ST, (**3**) and KPC-variant. (**4**) For further metadata and interactive exploration, follow this Microreact link: https://microreact.org/project/gUrL4jULY1JEPNExrFK4zH-globalcollection clonallineage258512.

distribution of carbapenemase genes differed between the countries, in clade 3 in Greece ST11 was associated with the $bla_{NDM-1}$ gene, in contrast, in Spain $bla_{OXA-48}$ as well as $bla_{OXA48-like}$ genes were present, whereas Serbian isolates in clades 1 and 2 carried either $bla_{NDM-1}$ or $bla_{OXA-48}$ (Fig. 5a). Overall, the isolates clustered largely by country and no transmission events were observed between countries based on the 21 SNP cutoff. For an in-depth study, we reconstructed a phylogenetic tree only with the Serbian isolates, due to most of them

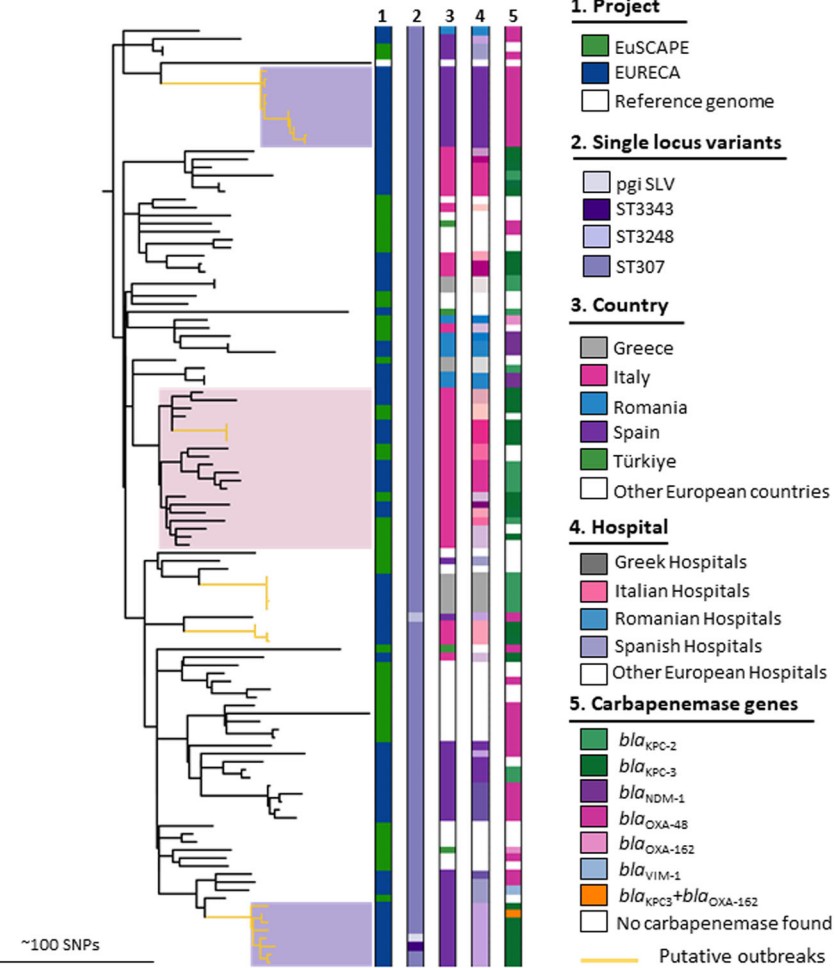

**Fig. 4 | Phylogenetic analysis of ST307.** Combined analysis of 117 isolates from EuSCAPE and EURECA. Reference genome CPKp1825 (WMHT01). Columns present the information about: **1** – Project, **2** – single locus variants, **3** – Country, **4** – Different hospitals (shades for each country), **5** – Carbapenemase genes. Putative outbreaks are highlighted with orange branches, and clades of interest discussed in the text are shaded with boxes (pink for Italy and purple for Spain).

belonging to two different STs to the founder ST11 (Fig. 5b). Interestingly, the ST340 clade contained predominantly carbapenem-non-susceptible EuSCAPE isolates without carbapenemase genes but with extended-spectrum beta-lactamase $bla_{CTX-M-15}$. In this clade, both highly clonal isolates within the same hospital (0-21 SNPs), and signs of a nationally spreading clone (22–100 SNPs for isolates collected from different hospitals) are visible. Isolates within the ST437 clade frequently carried either $bla_{NDM-1}$ or $bla_{OXA-48}$. Two of the subclades belonged exclusively to the EURECA collection and might represent a better approximation of the current circulating clones of carbapenem-non-susceptible ST437 in Serbia (Fig. 5b).

In a global contextualization of ST11 and variants using 1648 genomes publicly available in Pathogenwatch from different countries, the phylogenetic tree was clustered into several clades, with carbapenemase-producing EuSCAPE and EURECA ST11 isolates grouped in one particular clade (Supplementary Fig. 3b). Within this clade, there appears to be a particular subclade with $bla_{OXA-48}$ spreading in Spain, whereas the other European isolates carried $bla_{NDM-1}$. For isolates of ST437, the Serbian EURECA isolates clustered closely with isolates from Croatia and Slovenia, potentially signifying an Eastern European subclade (Supplementary Fig. 3c).

In ST101 ($n = 87$) (Supplementary Fig. 4a), EURECA isolates mainly came from Serbia ($n = 56$, 64%) and Romania ($n = 17$, 19%) and formed separate clades. We found that some of the Romanian isolates

appeared to be more diverse and were interspersed with isolates from Italy, Spain, and Türkiye, carrying different carbapenemase genes, however there was also a particular Romanian-only clade carrying $bla_{OXA-48}$ (Supplementary Fig. 4a, blue clade). In contrast, isolates from Serbia formed distinct clades, so we investigated them in more detail. To define the ancestral relationships of the ST101 in Serbia, we analyzed 135 genomes: 57 from EURECA collection, 42 from the EuSCAPE survey[8], and 36 genomes from Palmieri et al.[22]. The latter study recovered colistin resistance CRKP of ST101 between 2013 and 2017 and confirmed *mgrB* mutations as the major cause of colistin resistance in these isolates. They also reported $bla_{OXA-48}$ as the carbapenemase gene endemic in Serbia, which agrees with our results. The phylogenetic tree showed several clades, with the EuSCAPE isolates being the oldest and forming the most basal clade (Supplementary Fig. 4b). These isolates were also largely carbapenem-susceptible and did not harbor a carbapenemase (with one exception) (Supplementary Fig. 1b). There are further EuSCAPE isolates with either $bla_{OXA-48}$, $bla_{NDM-1}$ or a combination of both, however, the EURECA and Palmieri collections are intermingling in a separate, highly related clade, which was characterized by the presence of $bla_{OXA-48}$ (Supplementary Fig. 4b, shaded box). We found less than 21 SNPs between isolates of these two collections, and isolates had been obtained from a number of different hospitals, which might be indicative of a locally circulating clone. However, we cannot confirm this since we have no further information about the isolates of either

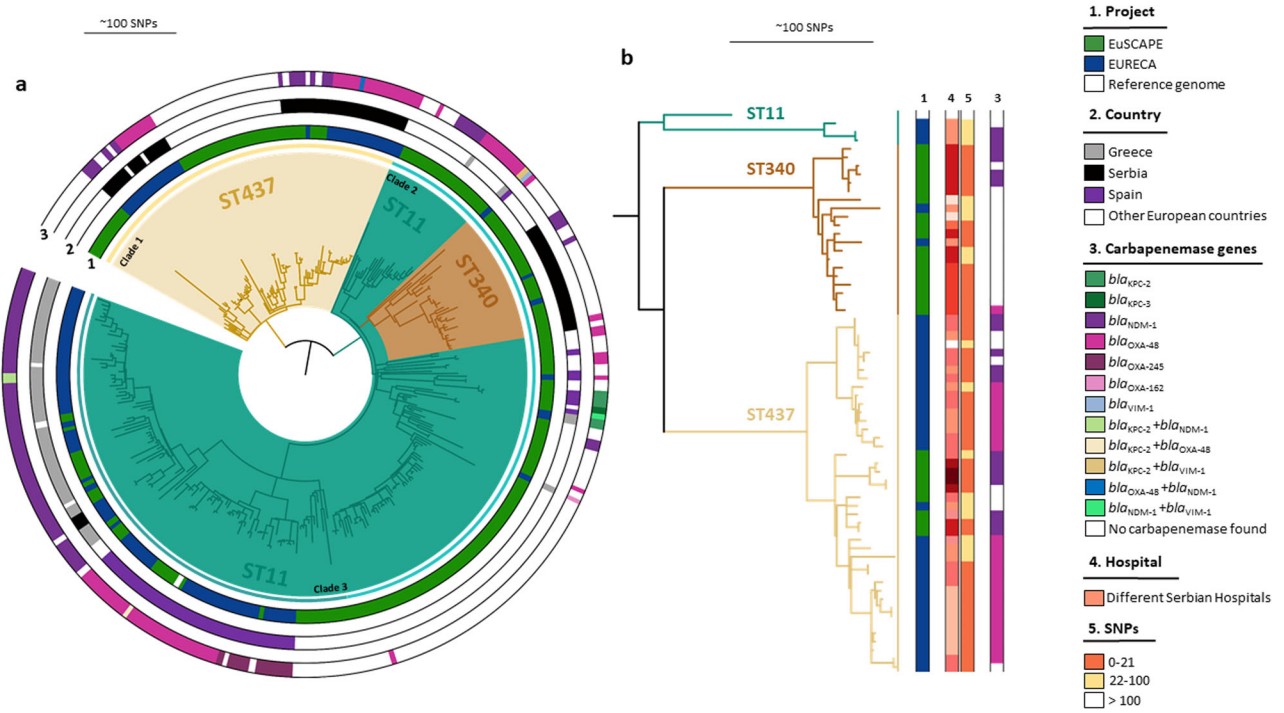

**Fig. 5 | Phylogenetic analysis of ST11. a** Combined analysis of 329 isolates from EuSCAPE and EURECA. Reference genome F64 (VILG01). Inner ring: project, (1) middle ring: geographic origin, (2) outer ring: carbapememase genes. (3)

**b** Phylogenetic relationship between isolates ST11 from Serbia only. Columns display the information about project, (1) different colors representing different hospitals in Serbia, (4) SNP distance, (5) and carbapenemase genes (3).

collection. Importantly, we can rule out that these isolates contributed twice to the different collections because the sampling time frames did not overlap.

### Other clonal lineages

Regarding other clonal lineages, one to highlight is the **ST147** ($n = 40$), which was mainly recovered from Spain ($n = 20$, 50%) and Greece ($n = 15$, 37%) in the EURECA collection. An outbreak in Tuscany, Italy, in 2018[23], and therefore we contextualized our isolates plus the EuSCAPE isolates (Supplementary Fig. 5). EuSCAPE and EURECA isolates were mostly diverse, with only two small clusters in Greece and Spain (EURECA collection) indicative of local clones (shaded boxes), whereas the Tuscany outbreak formed a separate, highly clonal expansion. Interestingly, we found six different carbapenemase genes within this ST, and their combination of two of these carbapenemase genes, even in the local lineages in Spain (isolates with $bla_{OXA-48}$ with and without $bla_{NDM}$) and Greece (isolates with $bla_{KPC-2}$ with and without $bla_{VIM}$), only the Tuscany outbreak was very homogenous in carrying the $bla_{NDM-1}$ gene.

The **ST15** was also diverse, recovered from a wide range of countries, and with different carbapenemase combinations (Supplementary Fig. 6). In Spain, we identified three clades (purple boxes); in two clades isolates carried $bla_{OXA-48}$ and the other, highly clonal clade with 14 isolates from the same hospital exhibited $bla_{OXA245}$ exclusively (EURECA collection).

Lastly, the **ST14** from the EURECA collection highlighted a local expansion with ST14 and the single variant ST2096 in Türkiye. Most of the isolates were from the same hospital except two, carrying the $bla_{OXA-232}$ and $bla_{OXA-48}$ gene on its own or in combination with $bla_{NDM-1}$ gene (Supplementary Fig. 7). Contextual analysis within a global collection showed that these isolates, especially ST2096, are part of a particular clade containing isolates from Türkiye, Saudi Arabia and a number of other countries (Supplementary Fig. 7; https://microreact.org/project/3mPo67JrFnL6Xt3cVSVrR1-st142096lineage globalcollectionproject).

### Discussion

Due to their rapid and efficient spread, CRKP is considered a public health problem worldwide. In order to monitor the clonal evolution and geographical expansion over time, new tools such as WGS have been implemented. The characterization of carbapenem-resistant isolates is essential for infection control purposes because of their impact on therapeutic decisions in clinics, hence laboratory detection providing high accuracy and fast diagnosis is key for accurate antibiotic delivery[24]. The entire collection of 683 isolates in this present study (COMBACTE-CARE EURECA) exhibited phenotypic resistance to at least one carbapenem commonly used to treat infections caused by multiple drug-resistant pathogens. The collection was split into 11 clonal lineages, the most prominent being ST258/512, ST11, ST101, and ST307, as the major high-risk clones distributed worldwide (Fig. 2a, b). The CRKp isolates in these 11 clonal lineages corresponded to 87% and 88% of the EuSCAPE and EURECA collections, respectively. In the EURECA collection, 27 singleton STs were found, with 150 found in EuSCAPE, as this collection encompasses the more diverse carbapenem susceptible population. As the EuSCAPE collection temporally preceded EURECA by two years, we contextualized and compared both collections to better illustrate the clonal distribution of CRKP in Europe. This showed that CSKP, only collected during EuSCAPE, is very diverse and contains many different STs, however, CRKP are grouped into the well-known high-risk clones circulating around the world[20] (Supplementary Table 1, Fig. 2). Some changes were observed in the prevalence in different countries: ST307 carrying 4 different carbapenemase genes increased in Italy, Romania and Spain, and a particular ST14 lineage was prominent in Türkiye. Differences were also observed with respect to carbapenemase genes: In Romania and Türkiye $bla_{NDM-1}$ was more prominent, in Spain more $bla_{KPC-like}$ genes were detected, and in Serbia $bla_{OXA-48}$ were found in a considerable proportion (Supplementary Table 1, Fig. 2).

Since the first report in 2001 of the clonal lineage ST258/512 in the USA associated with $bla_{KPC}$, its international spread has been rapid and extensive[8,14,15]. Using isolates belonging to ST258/512 of our current

collection along with previously published ones (Fig. 3), we confirmed the continental spill-over of ST258/512 into Europe from the USA, which began with several introductions into Greece and Italy, the expansion into Serbia and Montenegro, Romania and finally into Spain. EURECA helped to highlight more clearly the second introduction of this clone into Spain and Serbia (Fig. 3). We confirmed that this lineage linked to $bla_{KPC-like}$ is still prevalent in Italy, Greece, and Spain (Fig. 2)[25,26]. One isolate of the single locus variant ST1519 harbored $bla_{KPC-36}$ and was recovered from the blood of a hospitalized patient in Bologna, Italy, with a previous study reporting this novel $bla_{KPC-3}$ variant in the same city[27]. This new mutation confers resistance to ceftazidime/avibactam use for the treatment of CRKP[28].

ST258 was not the most abundant clone in other European countries such as Romania, Serbia, and Türkiye in the collections we compared (Fig. 2); this could be due to the rise of other high-risk clones like ST307, ST101, and ST11.

The emerging high-risk clone ST307 was first described in the mid-1990s, and it has since been reported worldwide with different carbapenemase resistance determinants[29–31]. This clone has also been responsible for nosocomial outbreaks in European countries such as Portugal, Spain, France, the UK, Germany, Netherlands, Slovenia, and Romania[32], and has been considered as a replacement for other high-risk clones, such as ST258 in countries like Colombia[33], South Africa[34], and Italy[35]. Furthermore, it was also responsible for nosocomial outbreaks harboring $bla_{KPC}$ variants conferring ceftazidime-avibactam resistance[24] and, more recently, cefiderocol resistance[36]. The presence of several chromosomal and plasmid-encoded factors associated with hypervirulence make ST307 a superior clone that can easily share large plasmids with other *Kp* and *Enterobacterales* species, such as the *Enterobacter cloacae* complex[31]. Indeed, the presence of a distinct capsule structure (rare in the *Kp* population) may contribute to the successful propagation of this lineage[37]. In the EuSCAPE and EURECA collections, we observed a regional circulation of this clone in Greece, Italy, and Spain (Fig. 4). In our two observed collections, the carbapenemase genes found in this clone often appeared to reflect the dominant carbapenemase in that country, namely $bla_{KPC-2}$, and $bla_{KPC-3}$ in Greece and Italy, $bla_{OXA-48}$ for the Spanish isolates. This may be a hint that the adaptability of ST307 with respect to carbapenemase uptake could be influenced by geographic prevalence, although interestingly, the Romanian ST307 isolates carried $bla_{NDM-1}$ (Fig. 4).

Large surveillance and sequencing efforts such as the current study not only improve our understanding of global clones but may also highlight regional circumstances that need to be taken into account in local infection control efforts. In Serbia, two different single locus variants of ST11 circulate: ST340, largely non-susceptible to carbapenems but without carbapenemases from the EuSCAPE collection, and ST437, mainly from the later EURECA collection, whose isolates often harbor either $bla_{OXA-48}$ or $bla_{NDM-1}$ (Fig. 5b). The founder ST11 is of minor importance as we found few isolates belonging to this ST, and putatively only signifies a single small introduction event from Greece (Fig. 5a). Comparing all non-susceptible isolates in EuSCAPE and EURECA from Serbia, the percentage of ST101 isolates has stayed the same, however the ST437 lineage has doubled in proportion. Since EURECA sampled fewer but completely overlapping hospitals with EuSCAPE, this may be an emerging trend amongst Serbian isolates. Also, in Serbia, a particular ST101 clone appears to have emerged, that is characterized by the presence of $bla_{OXA-48}$ (Supplementary Fig. 1a, b). From the current collections, we are not able to say whether the susceptible clones are still circulating in Serbia, or whether ST101 isolates would now exclusively be part of the non-susceptible clade. Similarly, in Spain, several different lineages of ST15 were co-circulating; two lineages were isolates carried $bla_{OXA-48}$ and one clade with $bla_{OXA-245}$ (Supplementary Fig. 3).

Regional differences exist in terms of the local dominant CRKP clones. In Greece, Italy, and Spain, ST258/512 was the dominant clone, while, in Serbia and Romania, ST101 was predominant, and ST14 was potentially expanding in Türkiye. The respective carbapenemase genes within these STs are likewise diverse. Overall, $bla_{KPC-like}$ was the most prevalent carbapenemase gene (46%) associated with the most abundant ST258/512. The second most frequent carbapenemase gene was $bla_{OXA-48}$ (39%), widely spread between different STs. Moreover, a relatively high proportion of isolates (5.5%) harbored two carbapenemases.

The combination of both EURECA and EuSCAPE collections helped to elucidate the high-risk clones circulating in Europe and the evolution of *Kp*. As both surveys had similar collection strategies, we were able to compare temporal changes from 2012–14 to 2016–18 through a comparison of dominant lineages, their associated carbapenem resistance genes, and the potential acquisitions of locally circulating genes. Collections such as these demonstrate that continuous surveys are necessary, and the sampling should always include susceptible background populations as well as highly resistant isolates in order to discern introduction from *de-novo* emergence.

These surveys also provide a blueprint for important European initiatives towards better planning for public health and infection control interventions.

## Methods

### Isolate collection and antimicrobial susceptibility
Isolates were recovered from patients diagnosed with bloodstream, intra-abdominal, pneumonia, and complicated urinary tract infection in 41 hospitals in nine countries (Albania, Croatia, Greece, Italy, Montenegro, Romania, Serbia, Spain, and Türkiye). The collection period was from May 2016 to November 2018. Overall, 49% of the samples were from blood ($n = 334$), 31% from urine ($n = 213$) and 20% from other sources ($n = 140$) (Supplementary Data 1). The countries with the major sample contributions to this study were Spain ($n = 202$), Greece ($n=174$), Italy ($n=111$), and Serbia ($n=110$). *Enterobacterales* isolates with MIC $\geq 1$ mg/L (dilution methods) or $\leq 22$ mm (disc diffusion, 10 µg disks) for meropenem or imipenem isolated from patients with the above infections were considered as putative CRE and studied; those producing carbapenemases and/or showing resistance to imipenem or meropenem according to EUCAST breakpoints were included. Collected isolates were phenotypically confirmed by disc diffusion at the EURECA central laboratory at the University of Antwerp. Further, the MIC values were determined by broth microdilution at SERMAS laboratory in the Hospital Ramon y Cajal in Madrid. The isolates were identified to the species level with MALDI-ToF, and then confirmed by whole genome sequencing.

### Whole genome sequencing and quality control
DNA extraction was performed using the Roche High Pure Template Preparation Kit. DNA concentration was measured with the Qubit fluorometer, followed by the sequencing library preparation using Nextera™ DNA Flex Library Prep (flow cell for 2x150bp paired-end sequencing).

Raw sequence reads were mapped to the reference genome MGH78578 (CP000647) using Smalt[38] and Samtools[39], with subsequent filtering with Genome Analysis Toolkit[40]. The minimum accepted average coverage was 30X per sample. The reads were de novo *assembled* using SPAdes v3.13.1[41] with Kmer sizes 21, 33, 55, 77, 99, 109, and 123. The expected size for the assemblies was between 5–7 Mb, 50–100 contigs, and N50 >100,000 bp. Kraken with mini-kraken database was used to check the species and potential contamination of samples[42]. Sequence types (ST) were determined using the multilocus sequences typing (MLST) software (https://github.com/tseemann/mlst). The average coverage was 50x with a genome size of 5.6 Mb and N50 of 259,285 bp. The median number of contigs was 86

(Supplementary Data 1). Sequencing data has been submitted to the ENA project PRJEB63349, with individual accession identifiers for reads and assemblies given in Supplementary Data 1.

Acquired antimicrobial resistance genes were identified using Abricate v0.9.8 (https://github.com/tseemann/abricate), with a local database based on ResFinder[43]. Capsular (K) serotypes were predicted by Kaptive, as implemented in the Kleborate genotyping pipeline (https://github.com/katholt/Kleborate)[44]. For variations of the $bla_{KPC}$ gene, we extracted the nucleotide sequence of all KPC-positive isolates. We then compared variations within the gene with known mutations associated with ceftazidime/avibactam resistance (GenBank accession number MH593787).

### Phylogenetic analyzes

Phylogenetic trees of the whole collection and subsets of particular sequence types were estimated using RAxML v8.2.12[45] based on SNP alignments after mapping to the reference genomes and removal of recombinant regions using Gubbins v2.4.1[46]. Reference genomes were chosen based on ST of interest with long-read sequencing and genome coverage >80X. Reference genomes for species overview: MGH78578 (GenBank accession number CP000647), ST11: F64 (VILG01), ST14: KPN528 (CP020856), ST15: P35 (CP053041), ST101: Kp_Goe_33208 (CP018447), ST147: HKP0064 (JACTAR01), ST258: 30660 (CP006923), ST307: CPKp1825 (WMHT01). Phylogenetic tree visualization was done with iToL[47]. Public datasets of interest (EuSCAPE[8], Italian collection of ST147[23], Serbian collection of ST101[22], and worldwide collection of ST258[13–16] were downloaded from the European Nucleotide Archive (ENA) and included in the analysis.

### Pairwise minimum SNP differences analysis

We used a PHP script based on the converted matrix from a DNA sequence alignment with snp-dist (https://github.com/tseemann/snp-dists) to determine the minimum SNP difference between pairs of isolates of the same ST. Based on their geographical origin, isolate pairs were analyzed in three different settings: different countries, the same country but different hospitals, and the same hospital. Plots were generated in R (v4.3.0)[48], using the ggplot2 (v3.4.4)[49] package.

### Global contextualization of lineages

We used the Pathogenwatch (https://pathogen.watch/) public genomes for a global contextualization of lineages. From the 48000 genome database associated with *K. pneumoniae*, we filtered by ST and created collections of up to 2000 genomes, corresponding to the maximum allowed.

Regardless of the geographical location, we excluded the samples that were highly clonal to include more diverse isolates, and we formatted the metadata in the same way we did for the EURECA collection. Microreact (https://microreact.org/) was used for interactive visualization.

### Reporting summary

Further information on research design is available in the Nature Portfolio Reporting Summary linked to this article.

## Data availability

Genome sequence data reported in this paper have been submitted to the ENA project PRJEB63349, with individual accession identifiers for reads and assemblies given in Supplementary Data 1. Reference genomes and genome data used from other studies are listed in Supplementary Data 3.

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

## Acknowledgements

The authors wish to thank the microbiologists from all participating laboratories in the EURECA study, and the members of the EURECA/WP1B group for their dedicated work and substantial contribution during the isolate collection phase of this study. The EURECA/WP1B group members are: Silva Tafaj (University Hospital of Lung Diseases 'Shefqet Ndroqi', Tirana, Albania), Arjana Tambic Andrasevic (University Hospital for Infectious Diseases, Zagreb, Croatia), Sophia Vourli (National and Kapodistrian University of Athens, Athens, Greece), Efthymia Protonotariou (General University Hospital of Thessaloniki 'Ahepa', Thessaloniki, Greece), Eleni Vagdatli (Hippokration General Hospital, Thessaloniki, Greece), Ioanna Voulgaridi (General Hospital of Larissa, Larissa, Greece), Iris Spiliopoulou (University of Patras, Patras, Greece), Ioannis Deliolanis (Laiko General Hospital, Athens, Greece), Maria Panopoulou (University Hospital of Alexandroupolis, Alexandroupoulis, Greece), Ergina Malli and Efthymia Petinaki (University Hospital of Larissa, Larissa, Greece), Efstathia Perivolioti (Evangelismos General Hospital of Athens, Athens, Greece), Theodora Biniari ("Agioi Anargyroi" General Oncology Hospital, Athens, Greece), Nicholas Legakis (Metropolitan General Hospital, Athens, Greece), Levantia Zachariadou (Aghia Sophia Children's Hospital, Athens, Greece), Anastassios Doudoulakakis (Panagiotis & Aglaia Kyriakou Children's Hospital, Athens, Greece), Vassiliki Pitiriga (Henry Dunant Hospital, Athens, Greece), Antonino Di Caro (National Institute for Infectious Diseases Lazzaro Spallanzani, Rome, Italy), Sara Giordana Rimoldi (Hospital Luigi Sacco, Milan, Italy), GianMaria Rossolini (Florence University Hospital, Florence, Italy), Maria Paola Landini (Policlinico Sant'Orsola Malpighi, Bologna, Italy), Teresa Spanu (Policlinico Universitario A. Gemelli, Rome, Italy), Milena Arghittù (IRCCS Fondazione Ca Granda Ospedale Maggiore Policlinico, Milan, Italy), Rossana Cavallo (Molinette Teaching Hospital, Torino, Italy), Anna

Marchese (University of Catania, Azienda Ospedaliero Universitaria "Policlinico-Vittorio Emanuele", Catania, Italy), Susanna Cuccurullo (University of Naples S.U.N./Monaldi Hospital, Naples, Italy), Paola Bernaschi (Ospedale Pediatrico Bambino Gesù, Rome), Roberto Bandettini (IRCCS Istituto Giannina Gaslini, Genova, Italy), Arsim Kurti (National Institute of Public Health of Kosova, Pristhina, Kosovo), Milena Lopicic (Clinical Center of Montenegro, Podgorica, Montenegro), Olivia Dorneanu (Clinical Hospital of Infectious Diseases of Iasi, Iasi, Romania), Mirela Flonta (Cluj Napoca Infectious disease Clinical Hospital, Cluj Napoca, Romania), Alma Kosa ("Dr. Victor Babes" Clinical Hospital of Infectious and Tropical Diseases, Bucharest, Romania), Daniela Talapan (The National Institute of Infectious Diseases Matei Bals, Bucharest, Romania), Mariana Buzea (Elias University emergency hospital, Bucharest, Romania), Camelia Ghita (Fundeni Clinical hospital Romania, Bucharest, Romania), Edit Székely (Spitalul Clinic Judetean de Urgenta Tg. Mures, Targu Mures, Romania), Snezana Jovanovic (Clinical Center Serbia, Belgrade, Serbia), Teodora Vitorovic (Clinical center of „Dragisa Misovic", Belgrade, Serbia), Deana Medic (Institute for Public Health Vojvodina, Novi Sad, Serbia), Branislava Kocic (Institute for Public Health, Nis, Serbia), Ana Perucica (Clinical Center "Zvezdara", Belgrade, Serbia), Begoña Palop-Borrás (Hospital Universitario Carlos Haya Málaga, Spain), Irene Gracia Ahufinger (Hospital Universitario Reina Sofia, Cordoba, Spain), Fe Tubau (Microbiology Department, Hospital Universitari de Bellvitge, Institut Investigacions Biomèdiques de Bellvitge (IDBELL), Barcelona, Spain and CIBER de Enfermedades Respiratorias (CIBERes), Institute de Salud Carlos III, Madrid, Spain), Fernando Chaves (Hospital Universitario 12 de Octubre, Madrid, Spain), Julio Garcia Rodriguez (Hospital Universitario La Paz, Madrid, Spain), Rafael Cantón (Servicio de Microbiología, Hospital Universitario Ramón y Cajal and Instituto Ramón y Caja de Investigación Sanitaria (IRYCIS), Madrid, Spain and CIBER de Enfermedades Infecciosas (CIBERINFEC), Institute de Salud Carlos III, Madrid, Spain), Francisca Portero (Hospital Universitario Puerta de Hierro Majadahonda, Madrid, Spain), Patricia Muñoz (Hospital Universitario Gregorio Marañón (SERMAS-HGM), Madrid, Spain), Cüneyt Özakin (Uludag University Turkey, Uludag, Türkiye), Banu Sancak (Hacettepe University School of Medicine, Ankara, Türkiye), Can Bicmen (Dr. Suat Seren Chest Diseases and Surgery Training Hospital, Izmir, Türkiye), Ufuk Hasdemir and Guner Soyletir (Marmara University School of Medicine, Istanbul, Turkey), and Zeynep Ceren Karahan (Ankara University Faculty of Medicine Department of Medical Microbiology). We thank Valentina Valenzuela Dallos, Sarah Klassen and Leonardo Duarte dos Santos for excellent technical support at the Medical Center – University of Freiburg, Germany. This research was funded by: Innovative Medicines Initiative Joint Undertaking (https://www.imi.europa.eu/) under Grant Agreement Nos. 115523 [COMBACTE-NET (Combatting Bacterial Resistance in Europe)] and 115620 (COMBACTE-CARE) to TK, JBF-A, MG-C, RC, HGo, JR-B, and HGr; Plan Nacional de I+D+I, Instituto de Salud Carlos III, Subdirección General de Redes y Centros de Investigación Cooperativa, Ministerio de Ciencia, Innovación y Universidades, Spanish Network for Research in Infectious Diseases (REIPI RD16/0016/0001 and RD16/0016/0011) and CIBERINFEC (21/13/00012, 21/13/00084) to JR-B and RC; JPI-AMR call (K-STaR 01KI1910) to HGr and SR; Federal Ministry of Education and Research (BMBF; TAPIR 01KI2018) to SR.

## Author contributions

M.B.S., S.A.M. and S.R. performed the bioinformatics analysis. S.R. supervised the bioinformatics work. T.K. and H.Go. were responsible for sample handling and phenotypic confirmation. J.B.F.-A., M.G.-C., R.C., J.R.-B. performed the MIC testing. R.C., H.Go., J.R.-B. and H.Gr. conceived the project and acquired funding.

## Funding

## Competing interests

The authors declare no competing interests.
