## [Peer Review File · Nature Communications]

International and regional spread of carbapenem-resistant *Klebsiella pneumoniae* in EuropeREVIEWER COMMENTS

Reviewer #1 (Remarks to the Author):

This is a large study of the genomes of carbapenemase-resistant *Klebsiella* in Southern Europe. The main results are a description of the clonal complexes and carbapenemase genes in these countries. One major finding is that there are regional variations in population structure and AMR genes.

The dataset is impressively large, documents a major collaborative endeavour (EURECA) and comprises a major body of work with collection, sequencing, and analysis. The genomes will be an incredible resource for researchers studying *Klebsiella*. The data has been released and a Microreact dataset is included, which is a credit to the authors for their approach to open access. The methods are very comprehensive (mapping to multiple references, good quality control), with no major obvious flaws and the manuscript was very clearly written and logical.

The in-depth analysis of the work is, however, underwhelming. The results are very descriptive and with little broader relevance outside of the Kp field. All figures (except for Figure 2) are different phylogenies of the data with/without additional public genomes, with traits (AMR, ST, country) mapped on to them.

Given these strengths and weaknesses, my only concerns are to do with the interpretation of the phylogenies in the results, with statements that may not be statistically robust.

Geographical analysis

There was a particularly large focus in the results on the phylogeography of different lineages but no quantitative assessment of this using formal phylogeographical models. For example, for CC258, the results postulate from the tips of the tree alone, introductions from the US into Europe and introductions from European countries to other European countries. Without a better understanding of the uncertainty of the ancestral geographical state in the nodes of the tree, it is difficult to say whether these stated results are robust and not misleading. It is very notable that the better sampled countries are the inferred geographical sources.

The manuscript would be improved by implementing the following with the individual clonal complexes:

Phylogeographic ancestral state reconstruction in a Maximum Likelihood or Bayesian framework

Subsampling of the geographical data to level the number of samples between countries e.g., up to 10 per country and repeat the results to make sure that these results are robust to sampling effort.

AMR analysis

In addition, the inference of the acquisition of the carbapenemase resistance genes suffers from the same problem. For example, the results state line 183 – “Then, ST238 acquired the blaKPC3 gene”. However, it seems equally likely based on the distribution at the tips that the blaKPC3 gene has been circulating in this clade since its emergence and instead switched to acquiring the blaKPC2 gene.

These results as stated are also not robust and completely open to interpretation. They could be taken out or strengthened with the addition of ancestral state reconstruction of the genes. Alternatively, is there any other information on the MGEs (i.e., genes/SNPs) that may additionally help to reconstruct the acquisition of these elements?

Comparison between EuSCAPE and EURECA

The results state that they compare the EURECA collection to the EuSCAPE collection (line 162-

163), but I could not find any comparison. This would have been welcome considering they are sampled over different times and could speak to whether different CCs change in prevalence over time. Instead, the collection is combined to give the picture of the different CCs in different countries. Figure 2 legend is even titled "Comparison of the ..." but there is simply no comparison (apart from 2a, which shows the different countries sampled) – the pie charts only show the combined datasets.

I think the manuscript would be improved by making this comparison i.e., for countries sampled from both datasets, having separate pie charts for the EURECA collection to the EuSCAPE collection. A formal statistical test of whether prevalences have changed would be beneficial, although I appreciate that there may be idiosyncrasies in data collection that could affect this result.

Minor comments

Lines 69-72 It is difficult to see how knowing the strain types and dominant circulating carbapenemase genes could have major relevance for designing treatment strategies. Would not most carbapenemase Kp would have the same treatment regime (with some exceptions as documented in Line 306-307)? Could the authors expand this explanation?

Could you please include in the method how the mutation in the blaKPC gene was characterized after BLAST extraction (line 128).

Phylogeny figure legends – could you please include which reference the data was mapped to. I appreciate that it is in the methods, but it has utility being here instead (or as well as).

Line 158. Just curious to know whether you believe that this was because they didn't assemble or that the phenotype may have a different genetic basis?

Line 347 "The shift in dominant lineages." I have missed this result entirely and it may be because there is little in the way of timescales, in either the trees or with the dataset comparisons. This should be taken out or expanded upon.

Line 353 infections should be infection.

Reviewer #2 (Remarks to the Author):

In the paper by Silva et al., the authors describe a collection of 683 CRKP genomes collected as part of the CONBACTE consortium from 2016-18, focusing on circulating clones of CRKP/carbapenemases and how these compare with the EuSCAPE study on 2013-14 CRKP in Europe. This is an interesting dataset and the authors have made some efforts to describe the diversity with clones/carbapenemases but I found the level of detail provided in the results often lacking (see specific comments below). For example, statements like 'clone X was prominent', 'regional differences in abundance' were often made without any numbers or percentages. Further, analyses around the clones could often be expanded upon; I didn't quite get a grasp of how many different clones / clone-carbapenemase combinations were detected across their entire dataset; how many of these were limited to particular countries vs others that have spread across multiple countries. For countries that overlap with the EuSCAPE collection, are there any differences over time regarding dominant clone or clone-carbapenemase combinations? Is there any evidence of the same circulating clones from 2013 (EuSCAPE study) through to 2018 i.e. do they cluster together? Related to this, it would be helpful to perhaps label their trees with clusters of interest to make the connection between the text and the figure clearer.

Specific comments:

* Authors mention that some isolates have multiple carbapenemases while others have no detectable carbapenemase, and don't further elaborate on these isolates? Were higher MICs observed for isolates that have multiple carbapenemases? Any ideas on the mechanisms behind strains that lack carbapenemase... the authors mentioned that they were re-cultured under

antibiotic selection - were there any MIC changes between the initial culture vs re-culture that would suggest lost of plasmid/carbapenemase during culture?

* Lines 131-141: How have the authors defined clonal complexes? In many studies, CC11 and CC258 are often considered part of the same lineage/clonal group; are the authors able to provide reasoning for why they have treated these as two separate groups? How were the references for each of the clones selected?

* Lines 162-173: How many genomes corresponded to CC258, CC11 and CC101? What were the regional differences? What do the authors mean by Greece had a higher diversity of clones? Need numbers to support statements like these. Were the same clone-carbapenemase combinations also observed in the EuSCAPE study? The authors mention OXA-48 carried on a promiscuous plasmid in various lineages; how have the authors defined same plasmid?

* Lines 176-193: This section was particularly difficult to follow. Is it perhaps better to use subclades to frame the number of unique circulating CC258 clones? For example, how many subclades do the EURECA genomes cluster into, and how many of these subclades include non-European isolates?

The figure contains other important results that have not been mentioned. For example, the majority of the CC258 isolates are ST258 or ST512. What about the KPC variants and how these associate with the various STs. How do ST-KPC allele combinations vary by country?

* Lines 182 - 183: Are these statements based on literature or what has been observed on the phylogeny. If the latter, I would use less 'definitive' language here, given that these are being inferred based off the available data.

* Line 196 - what do authors mean by 'adapting into a successful lineage' - ST307 has now caused multiple outbreaks in many geographies; should it therefore be considered a clone that has successfully established itself in healthcare settings?

* Line 199 - '...highest circulation of this clone...' - suggest rephrasing

* Lines 201-202; I don't quite follow this statement - isolates from the other countries also group into different clades? It is also unclear why the authors needed to construct two additional trees separating out isolates based on geography given that this information could be ascertain from the first tree containing all isolates?

* Lines 204 - so how many clusters of less than 21 SNPs were identified in total?

* Lines 206-207: were these all the same KPC allele?

* Lines 210-211: how many outbreaks? Is this based on the 21snp threshold? Quantify the number of isolates with oxa48 and kpc-3/oxa-162.

* Do the two ST11 subclades correspond to ST11 with distinct K loci? e.g. ST11 with KL47 vs KL64?

* Lines 217 to 219: When referring to the ST11 in Greece vs Spain, which of the two clades are they referring to? Do the ST11 from a particular country cluster together?

* Lines 232-233: Specify the number of isolates

* Lines 246-250: Might need rephrasing as I don't follow what the authors are trying to convey here.

* Lines 253-254: Provide some numbers please. How many CC147 isolates were there in total, and how many from Spain and Greece?

* Lines 258 - 260: What were the carbapenemases / how many distinct cluster-carbapenemase combinations were there?

* Lines 296-298: Are the authors able to provide some numbers here to clarify the statement. Is there any reason why France and Portugal is highlighted here? What about the other countries that have been sampled in this dataset where CC258 is not the dominant member?

* Line 319: Is NDM-1 not the dominant carbapenemase in Romania?

Minor comments

* Lines 34-36: What are the carbapenemases?

* Line 62: define WHO abbreviation

* Line 181; the authors state here light green isolates but don't include reference to Figure 3.

* Lines 176-193: consider revising some of the language used here as wording like 'expanded towards', 'visible in', 'showing fewer isolates', 'a clear introduction is now visible into Spain' reads very awkwardly.

* Line 192: what do the authors mean by 'looking for the new mutations'? Or do they mean allelic variants of KPC? Or novel mutations that haven't been previously described?

* Line 193: typo, ceftazidime

- * Lines 290-291: Missing citations to confirm that this lineage is still prevalent.
- * Line 584-585 (Figure 1 figure legend): the legend states that CC are labelled, but the labels in the tree state 'ST'
- * Line 598 (Figure 4 figure legend): typo, carbapenemase
- * Consider adding to Table 1 the number of different EURECA countries a particular clone was detected.

Reviewer #3 (Remarks to the Author):

The manuscript titled International and regional spread of carbapenem-resistant *Klebsiella pneumoniae* in Europe describes circulating high-risk clones and their evolution. It was very interesting to observe the dominance of CC258 clonal lineage and blaKPC-like carbapenemase gene. The findings highlight the importance of continuous monitoring of the spread of CRKP in the region and the implementation of infection control measures. The manuscript is well written. I give my consent to accept the manuscript for publication

REVIEWER COMMENTS

Reviewer #1 (Remarks to the Author):

This is a large study of the genomes of carbapenemase-resistant *Klebsiella* in Southern Europe. The main results are a description of the clonal complexes and carbapenemase genes in these countries. One major finding is that there are regional variations in population structure and AMR genes.

The dataset is impressively large, documents a major collaborative endeavour (EURECA) and comprises a major body of work with collection, sequencing, and analysis. The genomes will be an incredible resource for researchers studying *Klebsiella*. The data has been released and a Microreact dataset is included, which is a credit to the authors for their approach to open access. The methods are very comprehensive (mapping to multiple references, good quality control), with no major obvious flaws and the manuscript was very clearly written and logical.

The in-depth analysis of the work is, however, underwhelming. The results are very descriptive and with little broader relevance outside of the Kp field. All figures (except for Figure 2) are different phylogenies of the data with/without additional public genomes, with traits (AMR, ST, country) mapped on to them.

Given these strengths and weaknesses, my only concerns are to do with the interpretation of the phylogenies in the results, with statements that may not be statistically robust.

Geographical analysis

There was a particularly large focus in the results on the phylogeography of different lineages but no quantitative assessment of this using formal phylogeographical models. For example, for CC258, the results postulate from the tips of the tree alone, introductions from the US into Europe and introductions from European countries to other European countries. Without a better understanding of the uncertainty of the ancestral geographical state in the nodes of the tree, it is difficult to say whether these stated results are robust and not misleading. It is very notable that the better sampled countries are the inferred geographical sources.

We thank the reviewer for this comment. In fact, with respect to ST258 and its evolution, a large body of literature already exists as to when the clone was detected first and where. Hence, the ancestry of this particular clone can be followed even without phylogeographic models. We agree, that better sampled countries may be inferred as geographical sources, on the other hand, if cases only appear sporadic in other countries, this would indicate introduction events.

We have expanded the section on ST258 with the respective literature on this clone, to support the interpretation of the phylogeny (lines 141-175).

The manuscript would be improved by implementing the following with the individual clonal complexes:

Phylogeographic ancestral state reconstruction in a Maximum Likelihood or Bayesian framework
Subsampling of the geographical data to level the number of samples between countries e.g., up to 10 per country and repeat the results to make sure that these results are robust to sampling effort. Whilst this is a worthwhile idea, we don't think it appropriate with the current collection of samples. Subsampling to an even number of samples per country is difficult, since some countries contribute only few samples, and then the whole tree would be underrepresented. As we can see from the ST258/512 phylogeny as well, more samples with a higher diversity also match very well with the epidemiological observations and publications around the emergence of this particular clone, thus an ancestral state reconstruction would not add anything.

We included public genomes for some of the lineages, obtained from Pathogen.Watch. However, with these genomes, the sampling frame is different for each study. There might be a particular focus on a carbapenemase in a country or region, or an outbreak might be under investigation. This can

bias the number of samples and also the view on that lineage as a whole. We can therefore only use these phylogenies to pick out general trends or general observations (Supplementary figures 2, 3 and 7b).

AMR analysis

In addition, the inference of the acquisition of the carbapenemase resistance genes suffers from the same problem. For example, the results state line 183 – “Then, ST238 acquired the blaKPC3 gene”. However, it seems equally likely based on the distribution at the tips that the blaKPC3 gene has been circulating in this clade since its emergence and instead switched to acquiring the blaKPC2 gene. These results as stated are also not robust and completely open to interpretation. They could be taken out or strengthened with the addition of ancestral state reconstruction of the genes. Alternatively, is there any other information on the MGEs (i.e., genes/SNPs) that may additionally help to reconstruct the acquisition of these elements?

Upon re-examination of the tree, and the respective literature, we agree with the reviewer, and have amended the description of this (lines 143-151). The paraphyletic group carries KPC-3, with a switch in clade 1 to KPC-2 that also simultaneously changed the K locus. Clade 2 appears to emerge from the background of the paraphyletic group, and thus also carries KPC-3.

Unfortunately, with short read data, there is only limited analysis that can be done on the mobile elements carrying the antibiotic resistance genes. Generally, apart from ST258/512 where this has been corrected, we also do not imply ancestry of particular genes, rather pointing out their distribution.

Comparison between EuSCAPE and EURECA

The results state that they compare the EURECA collection to the EuSCAPE collection (line 162-163), but I could not find any comparison. This would have been welcome considering they are sampled over different times and could speak to whether different CCs change in prevalence over time. Instead, the collection is combined to give the picture of the different CCs in different countries. Figure 2 legend is even titled “Comparison of the ...” but there is simply no comparison (apart from 2a, which shows the different countries sampled) – the pie charts only show the combined datasets. I think the manuscript would be improved by making this comparison i.e., for countries sampled from both datasets, having separate pie charts for the EURECA collection to the EuSCAPE collection. A formal statistical test of whether prevalences have changed would be beneficial, although I appreciate that there may be idiosyncrasies in data collection that could affect this result.

We have now made this comparison clearer. We changed the pie charts (Figure2) and the section in the results (lines 104-131). Also, a table describing CRKP EuSCAPE isolates is included in Table 2. A statistical test might not prove useful given the small number of samples in some of the countries. We hope, that by reflecting how many samples were taken in the dominant countries, we can still convey the differences in prevalence sufficiently.

Minor comments

Lines 69-72 It is difficult to see how knowing the strain types and dominant circulating carbapenemase genes could have major relevance for designing treatment strategies. Would not most carbapenemase Kp would have the same treatment regime (with some exceptions as documented in Line 306-307)? Could the authors expand this explanation?

We agree, treatment regime would largely be the same. The main impact is on containment strategies and transmission control. We have removed the statement.

Could you please include in the method how the mutation in the blaKPC gene was characterized after BLAST extraction (line 128).

Please find this method in line 432-434.

Phylogeny figure legends – could you please include which reference the data was mapped to. I appreciate that it is in the methods, but it has utility being here instead (or as well as).

We thank the reviewer for their suggestion and have added this to our legends.

Line 158. Just curious to know whether you believe that this was because they didn't assemble or that the phenotype may have a different genetic basis?

We don't think that assembly is the problem. There was only one isolate where we only found a partial OXA-48 gene, in all other cases, no fragments of carbapenemases were found. Usually, in a given collection, some isolates do not turn out to be what they should be. This can have a number of reasons, and the main basis is that isolation, phenotyping, and sequencing are done in different places at different times from different cultures. In each of these steps, mistakes and changes to the culture can occur. Was there a carbapenemase initially? At the local site of isolation, only phenotypic tests were mandatory, not every lab performed PCR. Was there a carbapenemase upon receipt in Antwerp/Spain? Here, phenotypic testing was done using disk diffusion and combo disk test. It could be that there were undetected mixed strains, and that by chance a carbapenem susceptible isolate was picked for sequencing. However, even under antibiotic selection, we could not recover any carbapenemase, hence we have to conclude that in any of the previous steps, a carbapenemase either was not present or that the plasmid likely carrying it has been lost if antibiotic pressure has not been kept up. We have added a statement respective to this to lines 97-101.

Line 347 "The shift in dominant lineages." I have missed this result entirely and it may be because there is little in the way of timescales, in either the trees or with the dataset comparisons. This should be taken out or expanded upon.

We hope we have now clarified this point by expanding the sentence "As both surveys had similar collection strategies, we can compare temporal changes from 2012-14 to 2016-18 through comparison of dominant lineages [line 385-386]".

Line 353 infections should be infection.

Thank you, we have corrected this.

Reviewer #2 (Remarks to the Author):

In the paper by Silva et al., the authors describe a collection of 683 CRKP genomes collected as part of the CONBACTE consortium from 2016-18, focusing on circulating clones of CRKP/carbapenemases and how these compare with the EuSCAPE study on 2013-14 CRKP in Europe. This is an interesting dataset and the authors have made some efforts to describe the diversity with clones/carbapenemases but I found the level of detail provided in the results often lacking (see specific comments below). For example, statements like 'clone X was prominent', 'regional differences in abundance' were often made without any numbers or percentages. Further, analyses around the clones could often be expanded upon; I didn't quite get a grasp of how many different clones / clone-carbapenemase combinations were detected across their entire dataset; how many of these were limited to particular countries vs others that have spread across multiple countries. For countries that overlap with the EuSCAPE collection, are there any differences over time regarding dominant clone or clone-carbapenemase combinations? Is there any evidence of the same circulating clones from 2013 (EuSCAPE study) through to 2018 i.e. do they cluster together? Related to this, it would be helpful to perhaps label their trees with clusters of interest to make the connection between the text and the figure clearer.

We thank the reviewer for their feedback. As reviewer 1 had similar comments, we have now addressed the point of comparison with EuSCAPE by expanding and clarifying figure 2 (line 104-131). With respect to evidence of whether same circulating clones cluster together, this is what we presented in discussing several of the clonal lineages in more detail, always comparing EuSCAPE and EURECA. We didn't find any particular cluster that would implicate a transmission event, using the 21 SNP cutoff proposed by David et al 2019, apart from clusters within a single hospital in a given study. We have now added numbers and percentages, and have highlighted interesting subclades discussed in the text in the figures. We hope that this has made the manuscript clearer and more precise.

Specific comments:

* Authors mention that some isolates have multiple carbapenemases while others have no detectable carbapenemase, and don't further elaborate on these isolates? Were higher MICs observed for isolates that have multiple carbapenemases? Any ideas on the mechanisms behind strains that lack carbapenemase... the authors mentioned that they were re-cultured under antibiotic selection - were there any MIC changes between the initial culture vs re-culture that would suggest loss of plasmid/carbapenemase during culture?

We thank the reviewer for the comments and suggestions. The MICs values for isolates with multiple carbapenemases were similar to the isolates that have only one carbapenemase.

The MICs of the isolates under re-culture were not re-checked. We assume that the mechanism of resistance was on a plasmid, which has been lost in the time it was shipped to Freiburg. Please also see more detailed comment to reviewer 1.

* Lines 131-141: How have the authors defined clonal complexes? In many studies, CC11 and CC258 are often considered part of the same lineage/clonal group; are the authors able to provide reasoning for why they have treated these as two separate groups? How were the references for each of the clones selected?

We thank the reviewer for the questions. Indeed, Clonal Group CG258 encompasses ST11, ST340, ST437, ST258, ST512, and other minor single locus variants, based on the MLST relatedness. Given the very different behaviour and expansion of ST258/512 versus ST11 and the particular relevance of ST258, out of which ST512 (and variants) emerged, we prefer to treat these two lineages separately. We have now renamed the two groups as ST258/512 and ST11, to make this clearer, and have also amended the supplementary table accordingly.

We selected reference genomes based on ST of interest, and chose isolates with long read sequencing and high genome coverage >80X.

* Lines 162-173: How many genomes corresponded to CC258, CC11 and CC101? What were the regional differences? What do the authors mean by Greece had a higher diversity of clones? Need numbers to support statements like these. Were the same clone-carbapenemase combinations also observed in the EuSCAPE study? The authors mention OXA-48 carried on a promiscuous plasmid in various lineages; how have the authors defined same plasmid?

We kindly refer the reviewer to the previously included table 1, and we added more information in the figure 2 table 1 and Supplementary table 2, to clarify this information.

With respect to the OXA-48, as previous studies described the circulation of OXA-48 in IncL type plasmid, so far our analysis revealed (Plasmid finder) that the IncL plasmid is present in the same isolates that are carrying the blaOXA-48 gene. Long-read sequencing would be needed to investigate the plasmids further.

* Lines 176-193: This section was particularly difficult to follow. Is it perhaps better to use subclades to frame the number of unique circulating CC258 clones? For example, how many subclades do the EURECA genomes cluster into, and how many of these subclades include non-European isolates? The figure contains other important results that have not been mentioned. For example, the majority of the CC258 isolates are ST258 or ST512. What about the KPC variants and how these associate with the various STs. How do ST-KPC allele combinations vary by country?

We have expanded the discussion of ST258/512. Previously, clade 1 and clade 2 have been distinguished, and have been added to the tree, plus we have labelled the paraphyletic part of the tree. We hope that the section is now clearer to follow, we discuss the main findings reading the tree top to bottom (line 143-175; Figure 3).

* Lines 182 - 183: Are these statements based on literature or what has been observed on the phylogeny. If the latter, I would use less 'definitive' language here, given that these are being inferred based off the available data.

We have now added clearer references as to when the lineage was reported where, and refined our language.

* Line 196 - what do authors mean by 'adapting into a successful lineage' - ST307 has now caused multiple outbreaks in many geographies; should it therefore be considered a clone that has successfully established itself in healthcare settings?

The case of ST307 is a curious one, and we have tried to clarify this in the text. We agree with the reviewer that this lineage has indeed caused a number of outbreaks in various countries, however, from a global point of view (and we have now added Pathogen.Watch data on this), these outbreaks are varied in terms of the lineage and the carbapenemase. In contrast to ST258/512, there is not one or two single lineages that are spreading with particular characteristics. Hence, I am not sure whether this clone has "settled" on a particular strategy for success yet, or whether the strategy will be the adaptability to local circumstances (such as prevalence of a particular carbapenemase and the ability to pick that up). However, these outbreaks have rarely spread beyond a particular country, and are geographically refined. We have updated our discussion of this clone in this respect (line 181-215).

* Line 199 - '...highest circulation of this clone...' - suggest rephrasing
Corrected.

* Lines 201-202; I don't quite follow this statement - isolates from the other countries also group into different clades? It is also unclear why the authors needed to construct two additional trees separating out isolates based on geography given that this information could be ascertain from the first tree containing all isolates?

Thank you for your comments. We initially decided to reconstruct phylogenetic trees by particular countries, because particular clades by country were visible, however, we now agree that a single tree is actually sufficient given the resolution to discuss our findings (Figure4).

* Lines 204 - so how many clusters of less than 21 SNPs were identified in total?

Five subclades were identified with less than 21 SNPs (orange branches): two in Spain (highlighted in purple), two in Italy (the larger one highlighted in pink as it is part of a regional lineage), and one in Greece.

* Lines 206-207: were these all the same KPC allele?

The KPC producers harboured blaKPC-2 and blaKPC-3, detailed in the figure 4.

* Lines 210-211: how many outbreaks? Is this based on the 21snp threshold? Quantify the number of isolates with oxa48 and kpc-3/oxa-162.

We have added this information to the text, lines 181-184. 10 isolates had OXA-48, whereas 7 had kpc3 (including one which had both KPC-3 and OXA-162).

* Do the two ST11 subclades correspond to ST11 with distinct K loci? e.g. ST11 with KL47 vs KL64?

Yes, clade 3 corresponds to KL24, whereas clade 2 presents KL105 and KL15 (Supplementary table 3).

* Lines 217 to 219: When referring to the ST11 in Greece vs Spain, which of the two clades are they referring to? Do the ST11 from a particular country cluster together?

The isolates of ST11 from Greece and Spain were grouped in clade 3 (Figure 5a). Isolates largely clustered by country, as shown on the tree.

* Lines 232-233: Specify the number of isolates

We have added this information to the text (line 263-264).

* Lines 246-250: Might need rephrasing as I don't follow what the authors are trying to convey here.

We have tried to clarify this point. EURECA and Palmieri study isolates are highly related with SNP distances less than 21, and carrying the same carbapenemase gene, which may be an indication that this is a particular locally circulating clone in Serbia (line 263-268).

* Lines 253-254: Provide some numbers please. How many CC147 isolates were there in total, and how many from Spain and Greece?

We have added this information to the text.

* Lines 258 - 260: What were the carbapenemases / how many distinct cluster-carbapenemase combinations were there?

We found six different carbapenemase genes and their combinations, and within Spain and Greece (each with isolates less than 21 SNPs apart) all isolates carried one carbapenemase gene, sometimes in combination with another (line 280-284).

* Lines 296-298: Are the authors able to provide some numbers here to clarify the statement. Is there any reason why France and Portugal is highlighted here? What about the other countries that have been sampled in this dataset where CC258 is not the dominant member?

This paragraph has been changed.

* Line 319: Is NDM-1 not the dominant carbapenemase in Romania?

Please refer to figures 2e and f, and supplementary table S1. In Romania, 11/31 isolates had NDM-1 (and thus the proportion of NDM-1 was higher than in the EuSCAPE collection) but OXA-48 was present in 14/31 isolates, and thus dominant. This difference is small, and difficult to see in the figure.

Minor comments

* Lines 34-36: What are the carbapenemases?

We have added that information (line 28-30).

* Line 62: define WHO abbreviation

Corrected

* Line 181; the authors state here light green isolates but don't include reference to Figure 3.

Corrected

* Lines 176-193: consider revising some of the language used here as wording like 'expanded towards', 'visible in', 'showing fewer isolates', 'a clear introduction is now visible into Spain' reads very awkwardly.

We have reworded this paragraph extensively (line 141-177).

* Line 192: what do the authors mean by 'looking for the new mutations'? Or do they mean allelic variants of KPC? Or novel mutations that haven't been previously described?

Corrected. We searched allelic variants of blaKPC, and previously described mutations associated with ceftazidime/avibactam resistance (line 175-176).

* Line 193: typo, ceftazidime

Corrected

* Lines 290-291: Missing citations to confirm that this lineage is still prevalent.

Added

* Line 584-585 (Figure 1 figure legend): the legend states that CC are labelled, but the labels in the tree state 'ST'

Corrected.

* Line 598 (Figure 4 figure legend): typo, carbapenemase

Corrected.

* Consider adding to Table 1 the number of different EURECA countries a particular clone was detected.

Added.

Reviewer #3 (Remarks to the Author):

The manuscript titled International and regional spread of carbapenem-resistant *Klebsiella pneumoniae* in Europe describes circulating high-risk clones and their evolution. It was very interesting to observe the dominance of CC258 clonal lineage and blaKPC-like carbapenemase gene. The findings highlight the importance of continuous monitoring of the spread of CRKP in the region and the implementation of infection control measures. The manuscript is well written. I give my consent to accept the manuscript for publication.

We thank the reviewer for their kind remarks.

REVIEWERS' COMMENTS

Reviewer #1 (Remarks to the Author):

The authors have successfully addressed my concerns. While they chose not to complete my suggested phylogeographical analyses, they instead caution the language that they use, to only discuss the results that they have presented.

Reviewer #2 (Remarks to the Author):

In their revised manuscript, the authors have taken great effort to address concerns/remarks made in the first version, and this current version now includes in-depth phylogenetic/carbapenemase analysis for the high risk clones of interest. Importantly, these add interesting insights that help add context to their findings, something that was lacking in the first version. I have highlighted a few minor remarks to address as follows:

Line 36: suggest removing Enterobacterales to avoid confusion in the naming/nomenclature

Line 105: typo in 27.2%; replace comma with decimal point

Lines 107-108: would be good to state here in brackets what percentage of CRKp isolates do these 11 clonal groups account for in both collections

Line 154: awkward phrasing; do the author mean that the genomes from Serbia/Montenegro are nested within the clade comprising isolates from Greece?

Line 232: grammatical error; intermingled (or could use 'were interspersed')

Line 235: typo; plus symbol

Line 293: again, would be good to briefly quantify here how much of their CRKp isolates correspond to these high risk clones, and how many singleton STs were found

Lines 413-414: amend the wording here to say that capsular K types are predicted by Kaptive, implemented in the Kleborate genotyping pipeline

Line 421: what version of RAxML? Probably also need to include citation for this

Figure 1: The color font of the ST14 label is hard to see against the white background; could just use a slightly dark shade for the font label

Figure 2: increase font size of labels in the figure legend of panel D

Supplemental figure 2: The figure legend states 'Australian countries'; note that Australia is a single country. I would also suggest maybe using more contrasting colours for the data shown in the project column as it's difficult distinguish between the dark green versus blue against the purple. Indicate in the figure legend what the tip colours correspond to (country?)

Figure 3: The tips are presumably coloured by country of isolation? please indicate this on the figure legend

Reviewer #1 (Remarks to the Author):

The authors have successfully addressed my concerns. While they chose not to complete my suggested phylogeographical analyses, they instead caution the language that they use, to only discuss the results that they have presented.

Author response: We thank the reviewer very much again for their valued contributions to improve the manuscript.

Reviewer #2 (Remarks to the Author):

In their revised manuscript, the authors have taken great effort to address concerns/remarks made in the first version, and this current version now includes in-depth phylogenetic/carbapenemase analysis for the high risk clones of interest. Importantly, these add interesting insights that help add context to their findings, something that was lacking in the first version.

Thank you very much. We are glad the manuscript has now improved thanks to your comments.

I have highlighted a few minor remarks to address as follows:

Line 36: suggest removing Enterobacterales to avoid confusion in the naming/nomenclature
Done.

Line 105: typo in 27.2%; replace comma with decimal point

Thank you very much for spotting this; Germans confusingly use points and commas the opposite way round in numbers to the Anglophone convention.

Lines 107-108: would be good to state here in brackets what percentage of CRKp isolates do these 11 clonal groups account for in both collections

We have added those numbers.

Line 154: awkward phrasing; do the author mean that the genomes from Serbia/Montenegro are nested within the clade comprising isolates from Greece?

We have changed this sentence now to read "This introduction and expansion in Greece is detected in the EURECA collection, with isolates from Serbia and Montenegro nested within the diversity of the Greek isolates".

Line 232: grammatical error; intermingled (or could use 'were interspersed')

Done.

Line 235: typo; plus symbol

Done.

Line 293: again, would be good to briefly quantify here how much of their CRKp isolates correspond to these high risk clones, and how many singleton STs were found

We have added two sentences in this respect: "The CRKp isolates in these 11 clonal lineages corresponded to 87% and 88% of the EuSCAPE and EURECA collections, respectively. In the EURECA collection, 27 singleton STs were found, with 150 found in EuSCAPE as this collection encompasses the more diverse carbapenem susceptible population."

Lines 413-414: amend the wording here to say that capsular K types are predicted by Kaptive, implemented in the Kleborate genotyping pipeline

Done

Line 421: what version of RAxML? Probably also need to include citation for this

Done.

Figure 1: The color font of the ST14 label is hard to see against the white background; could just use a slightly dark shade for the font label

Done.

Figure 2: increase font size of labels in the figure legend of panel D

Done.

Supplemental figure 2: The figure legend states 'Australian countries'; note that Australia is a single country. I would also suggest maybe using more contrasting colours for the data shown in the project column as it's difficult distinguish between the dark green versus blue against the purple. Indicate in the figure legend what the tip colours correspond to (country?)

You are right, we have changed this. We have changed the colour shades; however, these are the colours we have kept all the way through the manuscript. The problem might be that there are few isolates of these collections in this figure, however, this can be explored more or better in the interactive MicroReact version.

Figure 3: The tips are presumably coloured by country of isolation? please indicate this on the figure legend

Yes, we have amended this.